# An Atlas for the Inkjet Printing of Large-Area Tactile Sensors

**DOI:** 10.3390/s22062332

**Published:** 2022-03-17

**Authors:** Giulia Baldini, Alessandro Albini, Perla Maiolino, Giorgio Cannata

**Affiliations:** 1Mechatronics and Automatic Control Laboratory, University of Genoa, 16145 Genova, Italy; giorgio.cannata@unige.it; 2Oxford Robotics Institute, Oxford OX2 6NN, UK; alessandro.albini@eng.ox.ac.uk (A.A.); perla.maiolino@eng.ox.ac.uk (P.M.)

**Keywords:** inkjet printing, large area, tactile sensors, robotic

## Abstract

This review aims to discuss the inkjet printing technique as a fabrication method for the development of large-area tactile sensors. The paper focuses on the manufacturing techniques and various system-level sensor design aspects related to the inkjet manufacturing processes. The goal is to assess how printed electronics simplify the fabrication process of tactile sensors with respect to conventional fabrication methods and how these contribute to overcoming the difficulties arising in the development of tactile sensors for real robot applications. To this aim, a comparative analysis among different inkjet printing technologies and processes is performed, including a quantitative analysis of the design parameters, such as the costs, processing times, sensor layout, and general system-level constraints. The goal of the survey is to provide a complete map of the state of the art of inkjet printing, focusing on the most effective topics for the implementation of large-area tactile sensors and a view of the most relevant open problems that should be addressed to improve the effectiveness of these processes.

## 1. Introduction

Tactile sensing has been studied for a long time, and a significant number of technologies have been studied in the literature [1,2,3]. Tactile sensors are used for sensing the location of contact and to measure surface properties, such as roughness, stiffness, and temperature. The research on tactile sensing has attracted intense interest in different fields, from biomedical engineering to industry. There are several applications of tactile-sensing systems, such as manual palpation, minimally invasive surgery, prosthetics, and sportswear as well as the agriculture, food processing, aerospace, and automobile industries [4]. Tactile sensing has also found with growing relevance in the robotics field, including robot manipulation and safe human–robot interactions [5,6,7,8,9,10,11,12].

In the domain of robot control, these operations have been treated using force/torque sensors [7,13,14,15,16,17,18]. Despite their accuracy, these do not allow dealing with complex physical interactions as in the case of multiple contacts or when major internal forces arise. In case of multiple contacts, only the resultant force would be detected, and therefore information on internal forces and contact locations are lost. Conversely, skin-like sensors used to process the distribution of tactile information, allow the reconstruction of the pressure distribution applied to the entire contact area [19,20,21,22,23,24,25,26,27].

Furthermore, inkjet printing has also become significant in recent years, and it can be considered one of the the most innovative methodologies for the manufacturing of large-area tactile sensors [28,29,30,31,32]. Inkjet printing is part of the class of additive manufacturing technologies [33]. The inkjet printing technique has all the advantages of the additive methodologies, in particular relating to the versatility and the cost-effective linked to a rational use of materials. In the inkjet printing process, a controlled print head deposits tiny droplets of ink on an underlying substrate [31].

This technology has emerged as a new solution for designing flexible and wearable sensors for its versatility with respect to other common technologies [1,28,34,35,36,37]. In fact, Computer Aided Design (CAD) tools allow straightforward geometric modification of the sensor layout as well as a precise control of the printer parameters, significantly reducing the setup time and costs paving the path for the customization of these products [28,38]. This is important in all applications where sensors customization is needed, and this is certainly the case for robot applications.

Furthermore, inkjet printing has a high versatility since it can be applied to different support materials allowing to print conductive patterns on different substrates made of both organic and inorganic materials, finally also reducing material waste [29,35,39,40,41,42,43]. Inkjet printing often takes place at room temperatures [44,45,46,47,48,49,50], and non-contact patterning with minimized contamination is involved [28,51].

Finally, inkjet printing leads to a limited number of manufacturing steps reducing the setup costs and time, thus leading to fast production and making large area sensors fabrication highly cost effective. [31,32,44]. The state of the art of tactile sensors has been discussed in the literature [2,52,53].

This paper aims to demonstrate the relevance of inkjet printing for the design and implementation of large-area tactile sensors with respect to the current state of the art (manufacturing techniques and technologies, including parameters, such as materials, layout, and processing). The contribution of the survey is to show how inkjet printing overcomes the main challenges arising during the design of tactile sensors and how this technology can boost the development of tactile sensors at large scales of production, leading to high technology readiness level (TRL) cost-effective devices ready for the market. Furthermore, a view of the most relevant open problems that should be addressed to improve the effectiveness of these processes is provided.

Figure 1 illustrates the topics addressed in this paper, which is organized as follows. Section 2 provides an overview of the requirements and major challenges arising in the design of large-area tactile sensors. In Section 3, the fundamental transduction principles used for tactile sensors are introduced. Section 4 describes the inkjet printing process with its advantages and shortcomings, highlighting the typical parameters with relevance to tactile sensors that can be achieved with this technology. Then, the main challenges associated with the inkjet printing process are discussed.

Section 5 gives a summary of inkjet-based sensor design solutions, including an assessment of the different characteristics of the fundamental transduction principles to show how inkjet printing allows overcoming certain difficulties that arise in the development of tactile sensors for real robot applications. In Section 6, we assess the most appropriate design solutions for different application areas. Open problems and conclusions are finally presented. The survey particularly focuses on the inkjet printing of large-area tactile sensors for robotics applications; however, all the treated topics can be extended to generic systems.

## 2. Requirements and Challenges in Tactile Sensor Design

In general, a tactile sensor is a device based on the transduction of a component due to induced contact stresses. Different transduction principles based on different techniques were used to measure contact information [54]. Each method has its pros and cons, and, depending on the applications, some methods are more suitable than others [55,56]. While tactile sensors are often built as matrices of sensitive interconnected elements (taxels), there is not a standard technology (in comparison to, e.g., image transducers) for the development of these devices.

Consequently, the overall system design can be linked to the type of application. In particular, for some applications, tactile sensing arrays built over large areas have been proposed. In this context, there are not specific numbers to define the requirements. Thus, we give the guidelines based on the task.

Examples of large area skins (in which information on the size (cm${}^{2}$) and number of sensitive elements can be found) are described in applications of *tactile-based control* (in the works of [27,57]) and *tactile-based recognition* (in the works of [58,59]). In *tactile-based control* tasks, the fundamental requirement is often the coverage: in these applications, the robot needs to be covered with tactile sensors since contact can occur anywhere.

Extended coverage allows for detection of the forces acting on the entire robot and to react accordingly. As a result, the spatial resolution plays a minor role, as we do not need to reconstruct the contact shape but to detect the lumped forces acting on robots. Conversely, in *tactile-based recognition* tasks, a high spatial resolution is required, since the aim is to recognize the spatial distribution of the acting force.

The development of tactile-sensing systems is affected by a large number of challenges and constraints [56,60]. One of these regards the fabrication of vertical vias in multi-layered tactile devices, in which functional devices from different layers need to be connected by vertical interconnects [61]. Some works addressed the problem by manually creating vertical feedthrough (creating holes and metallizing them), during or even after device fabrication [62,63]. This process is normally conducted in rigid double-layer devices.

Further difficulties arise in the case of multi-layered sensors (more than two layers), as the drilling must be done with a perfect cut to ensure that the subsequent metallization contacts all the conductive layers involved. Furthermore, reliable interfacial adhesion of flexible sensing elements with electronic components, such as wiring, integrated circuits, and other conventional devices, still remains a problem.

Since tactile sensors are devices subjected to significant mechanical stresses due to contacts, the problem of bonding and of sensor integration with the integrated electronics is an issue from the durability and robustness point of view.

Then, the choice of materials is an important constraint in the design of a tactile sensor. The ageing of the material used has to be considered as it leads to a decrease in the sensor lifetime and sensitivity. For large-area tactile sensors, the interconnection of the sensitive elements strongly depends on the geometry of the surface on which they must adapt. Therefore, other fundamental constraints arise during the design of tactile sensors. The sensor conformability is an important parameter since they need to mechanically adapt to a generic geometry, either planar or curved. Furthermore, sensors placement to ensure the maximum surface covering is needed, as well as simple calibration procedures.

Once interconnected, sensors need to communicate with each other via suitably routed wires (to not prevent, for example, the robot movement). As we discuss later, the key challenges that can be treated with printed electronics technologies mainly regard the system-level integration of tactile sensors (i.e., conformability, sensor distribution and placement, spatial calibration, and routing of wires) as shown in Figure 2, while the others still remain open problems. In this section, we describe how the system-level requirements have been addressed in the past.

### 2.1. Conformability

Tactile sensors should be conformable to cover arbitrary curved robot body parts easily. This is an important aspect especially when tactile sensors are distributed all over the body [60]. This goal has been pursued in the literature by implementing tactile sensors using flexible *printed circuit boards (PCBs)* and implementing discrete and interconnected tactile skin patches to improve the adaptability to the robot surface [12,64,65,66,67]. The available mechanically flexible tactile sensing solutions typically use PCB on rubber-like substrates and flexible PCBs as substrates, and the off-shelf sensing and electronic components are soldered.

Examples of such solutions are based on tactile skin modules having triangular, hexagonal or other basic shapes, interconnected to form tree or comb or general shaped patches. These techniques ensure good conformability but also have major drawbacks. In fact, sometimes, the bending radius of flexible PCBs is large, and such solutions are only suitable for body parts, such as the arms of a humanoid.

Furthermore, skin patches make the sensors easily adaptable to different surfaces; however, as we discuss in the next paragraph, they lead to surface covering problems. Finally, the flexibility of a sensing structures does not always ensure conformability in all directions.

Many available structures can bend only along one axis and hence can conform to surfaces as cylinders, which approximates the arm of a robot; in this context, many simple flexible PCBs fail. One way to address this issue could be based on the use of intrinsically stretchable sensors. As discussed in [60], this poses new challenges in introducing new materials and manufacturing processes.

Bringing these ideas to the extreme, the concept of stretchability could also be proved from the electronic components level by manufacturing stretchable electronic devices. The realization of stretchable devices with traditional materials, such as silicon is a challenge owing to their intrinsic properties, such as brittleness that limit their ability to stretch or bend [68,69].

Furthermore, metals, such as gold and copper, have been preferred for interconnects and electrodes, owing to their high electrical and thermal conductivity that permits an influx of large current and fast transmission of signals [68]. However, when it comes to flexibility, the metals were found to have limited use as they are not sufficiently elastic [68].

### 2.2. Sensor Distribution and Placement

A surface covered with tactile sensors often has a complex geometry characterized by variable curvatures. Hence, the need to create customized sensors has emerged. The spatial distribution and placement is an essential challenge because the number of tactile sensors and their distribution across the sensing surface affects the quality of measurements and the effectiveness of tactile data utilization.

Appropriate placement of the sensors can ensure the maximum surface covering, guaranteeing appropriate sensitivity and taxel distribution over large areas allowing to measure multiple touches over the whole surface of the device [70]. The state of the art of tactile technology does not allow the fabrication of sensors, which can be easily adapted to arbitrary geometries.

Custom solutions can be complex, time consuming, and expensive since they might require a complete redesign of the manufacturing process. Approximate covering solutions were proposed to ensure sub-optimal placement of general-purpose tactile skin patches [56]. Examples of such solutions include tactile skin patches with triangular [64,65], hexagonal [66], and tree- or comb-shaped modules [12,67].

These types of structures start from simple shapes, and they are further extended to approximate complex geometries. For example, the triangles are inspired by computer graphics for creating complex solids. For this reason, various large-area tactile sensors are constructed as meshes of triangles or hexagons to simplify the approximate placement of the sensors on general robot body geometries. Once shapes have been obtained, the interconnections can be done by software that solves the optimization problem to generate the mesh.

Preliminary works by Anghinolfi et al. [71], further extended by Wei et al. [72] and Mastrogiovanni et al. [73] have discussed the possibility of obtaining suboptimal covering of surfaces based on standard robot skin patches geometries, typically formed by triangular or hexagonal meshes.

Software methods can be either separate or complementary to the hardware part, as the aforementioned geometries do not solve the problem by themselves, especially for their physical limits (if the sensor is bent beyond a certain limit, it breaks). Thus, the challenge is the quick design of ad hoc components.

### 2.3. Spatial Calibration

As previously discussed, currently, sensor patches are placed on the surface to be sensorized with significant uncertainty in the displacement of the tactile elements. This requires time-consuming geometrical calibration procedures to obtain specific calibration files [74,75,76,77,78,79]. These procedures are tedious and may require specific equipment.

### 2.4. Wiring

The wiring of large-area tactile sensors is one of the most critical problems related to the development of tactile sensors [80,81]. In robotics applications, the large number of wires bring to the fore such problems as routing cables through the robot’s joints and optimizing the use of the limited available space in the robotic shell, due to existing motors and electronic components [56].

Available wireless technologies [82,83] can not be used for robotics applications because they are not reliable and they are not deterministic, and thus problems on robots control arise [80]. In general, as the number of sensors increases, so is the number of wires needed to address elements, acquire and transmit the data, and power the sensors. As reported in [80], the number of wires has an inverse relation with dexterity and a direct relation with the time needed to scan a set of taxels or array. Fewer wires call for the serial access of data, which is slower than parallel access that requires a large number of wires [80].

Thus, the challenge is related to how many wires have to be integrated. If many wires are involved, the problem of routing arises; however, if there are few wires with a serial access, the bus generates band saturation. In order to find optimal routes and avoid wire collisions, Wasserfall et al. [84] proposed an algorithm solution for local, topology-aware wire routing.

## 3. Tactile Sensing Principles

This section introduces the fundamental transduction principles of tactile sensors. In the literature a huge number of transduction principles have been proposed [1,3,52,55,85,86]. Piezoresistive, capacitive, piezoelectric, optical, inductive, and magnetic methods have often been the preferred choice of sensor designers. Although other solutions are emerging (e.g., triboelectric transducers [52,55,86]), there are still not large-area demonstrators that show the applicability of the technology in high TRL tasks (e.g., TRL 4 or TRL 5). In this section, we give a brief review of the main transduction methods and their relative advantages and disadvantages.

### 3.1. Piezoresistive

The mechanism of piezoresistive sensors is based on the transduction of pressure into the variation of resistance. The resistance variation of piezoresistive sensors mainly comes from the contact resistance between two conductive electrodes subject to an external pressure. The voltage–current characteristic of a simple resistive element can be expressed as, V=IR; where *V* is the voltage, *I* is the current, and *R* is the electric resistance of the material. A standard piezoresistive sensor is composed of a thin film of material applied to a plastic substrate.

Piezoresistive sensors have been widely studied for their simple fabrication, structure, low cost and easy signal acquisition. They allow transducing pressure input stimuli starting from zero frequency (constant pressure) up to frequencies of the order of kHz. Piezoresistive sensing is less susceptible to noise; however, it is affected by hysteresis, leading to lower frequency response compared to capacitive stactile sensors, and they can only be used for dynamic measurements with limited spatial resolution [4]. The reported maximum piezoresistive sensing sensitivity can reach 0.25 mV/nm [87].

### 3.2. Capacitive

Capacitive tactile sensors are based on the change in capacity by mechanically changing the geometry of the capacitor. The capacitive pressure sensors are mainly based on the parallel plate capacitor, and the capacitance is determined by the equation: $C=\frac{\epsilon_{0}\epsilon_{r}A}{d}$, where $\epsilon_{0},\epsilon_{r}$ are, respectively, the vacuum electric permeability and the dielectric constant of the material used. *A* represents the electrode area and *d* is the distance between the electrodes. When applying pressure perpendicularly to the electrodes, *d* changes leading to the variation of capacitance, whereas in some design also *A* can change as the effect of shear force components [55].

Parallel plate capacitors are the fundamental structure for capacitive sensing. Rectangular stripes, pyramidic structures, spheres, and pillars are all other structures used [1]. Capacitive touch sensors have been well received for their low power consumption, high sensitivity, high repeatability, and simple device construction [88]. They generally have a high spatial resolution and large dynamic ranges; however, they might be susceptible to multiple types of noises [4].

Furthermore, as for piezoresistive sensors, capacitive sensors can measure input stimuli starting from constant pressures (zero frequency) up to frequencies limited by the required output sampling frequency and by the acquisition electronics. The best performance of capacitive tactile sensors ever reported is a minimum resolution of 3 Pa with a sensitivity of 0.55 kPa${}^{-1}$ [89].

### 3.3. Piezoelectric

Piezoelectric devices produce an electric charge proportional to the mechanical stress they are subject to. A typical readout mechanism is based on the usage of a charge amplifier leading to an output voltage, which is proportional to the mechanical stress. The sensing principle leads to sandwich configurations as the general sensing structures for piezoelectric tactile sensors, where piezoelectric layers are deposited between two electrode layers. As with the capacitive tactile sensors, convex structures, such as mesas and spheres, have been integrated as a contact promoter.

The peculiar characteristic of piezoelectric transducers is that they cover wide measuring ranges. The maximum measurable pressure range was reported as 100 MPa [90]. Thus, it is possible to use the same sensor to measure very small or very large forces. Furthermore, piezoelectric tactile sensors are attractive due to their fast response, high dynamic sensitivity and low material costs. They exhibits very high-frequency response, making them the best choice for dynamic signal sensing [4]. Unlike capacitive or piezoresistive tactile sensors, piezoelectric tactile sensors can not transduce the effect of constant input stimuli, i.e., constant pressures (zero frequency input).

### 3.4. Optical

Optical tactile sensing is implemented by coupling geometric change of electromagnetic waveguide with the modulation of the wavelength, phase, polarization, or intensity of the wave. Usually transduction occurs when changes in the tactile medium modulate the transmission or reflectance intensity, or the spectrum of the source light, as the applied force varies.

Key sensing structures for tactile sensing are optic fibres vertically placed towards the sensing surface and waveguides sandwiched between substrate and contact interface structures [1]. Optical tactile sensing is immune to common lower frequency electromagnetic interference generated by electrical systems, which is its major advantage. Optical sensors generally have high spatial resolution and wide dynamic response range.

As reported in [1], optical tactile sensing can be used for sensing surface roughness, compliance, and shear and vertical stress. For roughness measurements, the reported best resolution is around the 100 nm level [91]. For stress and mechanical forces, a resolution of 0.02 N was reported for optical sensors used for minimally invasive surgery [92]. Optical tactile sensing has shown great potential in applications requiring flexibility and portability [93,94]. Although they have many benefits, their size and rigidity are the major disadvantages [4].

### 3.5. Inductive and Magnetic

In inductive and magnetic tactile sensors, a primary coil induces a magnetic field, which is sensed in a secondary sense coil. Modulating the mutual inductance between the coils, for example by changing the length of an iron core in the case of a linear variable differential transformers, in turn modulates the amplitude and phase of the voltage measured in the sense coil. These sensors have a high dynamic range and an often rugged construction but are bulky in size, which leads to a low spatial resolution when arrayed.

Due to their mechanical nature, they have low repeatability as coils do not always return to the same position between readings. Since these sensors use an alternating current in the primary coil, hence producing an output voltage at the same frequency, they require more complex electronics than normal resistive tactile sensors as the alternating signal amplitude must be demodulated [4].

## 4. Inkjet Printing Technology

### 4.1. Inkjet Printing Modes of Operation

The use of the inkjet printing technology allows a calibrate use of materials, making possible on one hand the versatility in the construction of components, and on the other hand a targeted use of materials with reduction of waste and, therefore, benefits in terms of costs and environment.

There are two modes of operation for inkjet printing: *drop-on-demand (DoD)* and *continuous inkjet printing (CIJ)* as shown in Figure 3. In most DoD printers, the droplets are induced by a piezoelectric actuator. In CIJ printers, a continuous electro-conductive stream of fluid is delivered through a nozzle because of piezoelectric actuator vibrations, regulating the breakup of the stream into individual uniform droplets with uniform spacing. In the DoD method, single drops of ink are released onto a wide range of substrates in response to a digital signal or waveform, allowing for the printing of both aqueous and UV curable inks.

This allows for the control and calibration of the ejection of ink droplets from each nozzle. This method is widely used due to its high placement accuracy, controllability and efficient material usage, where no excess ink is wasted, as it is only dropped where and when needed. As a result, this method is frequently implemented in research and in industrial applications. Furthermore, it is adaptable to the printing of water-based inks containing conjugated polymers, as it uses lower temperatures.

Piezoelectric inkjet printing relies on the mechanical action of a piezoelectric membrane to generate a pulse. Nozzle sizes are in the range of 20–30 mm, and droplets sizes are in the range of 10–20 pL. The diameter of an ink drop on a substrate is about twice the size of the released drop [95,96,97].

Continuous inkjet technology differs from DoD in the way the ink is deposited from the cartridge. Specifically, CIJ produces a continuous stream of ink, where drops break up from the print head nozzle by applying harmonic modulation. The separation of the drops is determined by the modulation frequency and the speed of the jet. Continuous inkjet systems typically produce 80 to 100 mm droplets travelling at speeds of 20 m/s with drop frequencies that can exceed 250 kHz.

In the most common implementation of CIJ printing, electrostatic charging and deflection are used to select and steer individual drops to define the final printed pattern. On the other hand, CIJ can operate with fluids of lower viscosity and at a higher drop velocity than DoD. However, CIJ printing requires a larger amount of fluid. This demonstrates the advantage of DoD systems over CIJ in the area of inkjet-printed electronics as it consumes less ink and accordingly reduces the overall fabrication cost [44,51].

### 4.2. Inkjet Printing Components

In any inkjet printing process, three main components are required: the inkjet printer, the ink, and the substrate, and these are shown in Figure 4. There are several commercially available inkjet printers that are widely used for research purposes. Among these, Dimatix Fujifilm printers (DMP-2831 and the newer DMP-2850) are the most commonly used types of inkjet printers for research [28]. Industrial devices (from Fujifilm) are based on the same technology.

Inkjet printing technologies have been widely used for the development of flexible electronics as they offer a fabrication alternative to lithography-based approaches [68,98], introduce the use of new intrinsically stretchable inks [39,97,99,100,101,102,103,104,105,106,107,108,109,110,111,112,113,114,115,116,117,118,119,120,121,122,123,124,125,126,127,128,129,130,131,132,133], and avoid the challenges related to the brittleness of traditional materials used to manufacture tactile sensors [7,82,134,135,136,137,138,139,140,141,142,143].

Furthermore, inkjet technology makes the system conformable by printing on different types of flexible substrates [28]. Paper substrates are the most common flexible type of substrates, which are used in inkjet printing of electronics and sensors, as will be discussed in Section 5.2. Polymer substrates, such as polyether imide, polycarbonate, polyarylate, polyamide, polyethylene, and terephthalate are also types of commonly used flexible substrates [126,128,144]. PDMS is frequently used as a stretchable substrate [98,145,146,147].

This is a silicon-containing elastomer, which is also widely used to construct microfluidic devices and lab-on-chip systems. On one hand, the aforementioned stretchable inks and flexible substrates allow overcoming the challenges related to the conformability of the tactile sensors. On the other hand, these two main components allow producing printable sensors with printable stretchable wires as in the work of Albrecht et al. [148], which may find use in artificial skin for humanoids.

The optimized wiring is one main advantage of the application of inkjet printing for tactile sensors, which leads to the printing of wires instead of physically turning them, for example, between the robot joints. Thus, inkjet printing introduces a difference between printed wires and non-printed ones. In the first case, the stretchability of wires permits a simpler routing, allowing a high level of integration to the entire system. In the second case, the problem of flying and non-stretchable cables arises, leading to the routing of wire problems.

### 4.3. Relevant Parameters

The resolution of inkjet printing is an important parameter that mainly depends on the ink volume propelled by the inkjet-printing machine. The width of the printed tracks is limited by the resolution of the printer. For example, the resolution of Dimatix is 10–30 μm depending on the drop volume [31]. The constituents of the ink, the substrate, the sintering process, and the voltage waveform applied to the jetting nozzles affect the resolution and accuracy of the inkjet-printing process [31]. The best resolution achieved using inkjet-printing technology was 2 μm, and ithis was proposed in the works of [149,150].

### 4.4. Challenges of Inkjet Printed Tactile Sensors

This section analyses the major challenges related to the manufacturing of inkjet-printed tactile sensors. The first problem regards the choice of the ink, looking at its proprieties, such as the viscosity, surface tension, and contaminants or polymer chains. Other issues include the adhesion of the inks to the selected substrate, sintering processes after the printing and the sensor’s mechanical durability.

When designing inks for inkjet printing, the properties of interest are primarily the viscosity and surface tension [151]. Thus, inks must have the required value of viscosity as not to clog the printer nozzles [51,152,153] and a required value of surface tension to permit the bonding between the drop and the substrate [86]. In the case of too viscous materials, a dilution with a solvent is required, until the mixture reaches the right viscosity [151,154]. Here, the challenge is to choose the right solvent concentration to maximise the ink’s material-content.

An excess of solvent with respect to the main material can alter the properties of the latter. Furthermore, the need for a right viscosity upon the extrusion through the printhead can also limit the range of materials that the inkjet printing technique can use. Once the appropriate dilution is found, the ink must pass through a filter before being inserted in the cartridge. Filtering is an important step to prevent nozzle clogging prior to printing, either from any contaminants, or from polymer chains or other ink components that are too large [151].

Fluid particles must generally be 1/100 of the printer nozzle size [28]. For example, the 10 pL cartridge of the commercial printer Fujfilm Dimatix Material Printer DMP-2850 has a nozzle of 21 μm; in this case, the particle diameter must be 0.2 μm, and thus a filter with a pore size of 0.2 μm needs to be used.

The second key issue is the adhesion of the ink to the selected substrate. In several works, the substrate surface is modified using plasma treatments to make it hydrophilic, thus, allowing the adhesion of ink particles [96,155,156,157,158,159,160,161]. Adhesion problems can also arise in the case of multiple depositions. This challenge is particularly sensitive in the case of PDMS-based inks [151]. In this example, the problem is the adhesion of the silver nanoparticle (AG NP) ink on top of PDMS layers.

The authors proposed the use of nitrogen plasma treatments for printing the conductive layers, enhancing its adhesion with the hydrophobic PDMS. Mikkonen et al. [162] studied and compared other methods to improve the adhesion between the ink and the substrate. These methods include other chemical modifications, such as (3-mercaptopropyl)trimethoxysilane (MPTMS) and pyrolytic coatings.

Another method to improve the adhesion is ink sintering, which also has a decisive role regarding the ink conductivity properties [28]. Sintering is the last step in an inkjet printing process, and this is essential for the evaporation of the solvent that might be present in the ink. This is performed by applying heat to the printed pattern. On the one hand, high sintering temperatures increase electric conductivity by dissolving more of the solvent and further fusing the nanoparticles of the conductor [28]. On the other hand, high sintering temperatures may harm the substrate and change its properties [28].

In fact, sintering processes ensure high conductivity; however, they can also reduce the lifetime of the substrates [31]. A third key element in the development of inkjet-printed tactile sensors is their mechanical durability [28]. Clearly, a tactile sensor that can sustain prolonged bending or stretching, while maintaining its response characteristics is desirable. As a matter of fact, electrical conductivity of the sensors electrodes is the priority.

Typically, sensors are tested for the number of bending/stretching cycles they can undergo before cracking and/or slipping of the conductive lines [28]. In the case of sintering process, the solvent in the deposited ink evaporates and the conductive lines, which have different thermal expansion coefficients with respect to the substrate, tend to crack, thereby, affecting the electrical conductivity of the printed lines [163].

However, various strategies have been adopted to reduce this effect. One possible strategy is to print thicker conductive tracks (even if thicker tracks may reduce bendability). Alternatively, using thinner substrates. Otherwise, using flexible inks and substrates [28]. Furthermore, the strategy of sandwiching the conductive lines between two elastomer layers has shown a reduction in crack formation [151,164]. During sensor life cycle tests, slipping of the printed conductive lines with respect to the substrate can occur if the adhesion between them is insufficient [164]. In this context, the challenge can be overcome using the right adhesion process (as discussed above).

## 5. Inkjet Printed Tactile Sensors

For inkjet-printed tactile sensors, there are three types of transduction technology: piezoresistive, capacitive, and piezoelectric and, depending on the application, some are more suitable than others [55,56], as will be discussed later.

A summary of inkjet-based sensor design solutions is described in the following, including an assessment of the different characteristics of the various manufacturing solutions.

### 5.1. Piezoresistive

Using inkjet printing technology, silver, single-wall carbon nanotubes (SwCNT) and graphene are conductive materials used in many works [101,132,139,165,166]. The contribution of inkjet printing for the manufacturing of these kind of sensors is to introduce a cheaper alternative aiming to minimize the complexity without compromising the quality and sensing performance for the target application.

Significant advantages of these aspects are discussed in the work of Lo et al. [139]. Here, an inkjet-printed resistive pressure sensor that offers high sensitivity and can be fabricated using a very simple process was reported. The device is comprises a conductive silver nanoparticle (AgNP) layer directly printed onto a polydimethylsiloxane (PDMS) substrate and encapsulated by a Very High Bond (VHB) tape. The pressure is measured by changing electrical resistance caused by pressure-induced strain in the printed AgNP thin film.

From the geometry point of view, Li et al. [101] proposed a novel ring-shaped piezoresistive pressure. Carbon nanotube (CNT)-polyimide (PI) nanocomposites with different CNT concentrations are prepared in a liquid mixture as the fabrication material. The ring-shaped nanocomposite thin film device is deposited on a circular polyimide diaphragm by inkjet printing fabrication method. The drop-on-demand fabrication process of the inkjet printer enables the fabrication of nanocomposite thin film in various geometries and shapes to achieve better structural compatibility with the sensor port and over-all measurement accuracy.

Due to the last one sensor features, the sensors offer some advantages as an easy fabrication process, excellent flexibility, and high sensitivity. Figure 5 shows the piezoresistive tactile sensors manufactured in [101,139].

### 5.2. Capacitive

Inkjet printing technology facilitates the utilization of a variety of flexible substrates for various designs [31]. Polyethylene terephthalate (PET) is widely used as a flexible substrate for inkjet-printed capacitive sensors [100,129,137,142]. Furthermore, printed electronics on paper allows the implementation of flexible sensors, and it has been used to manufactured printed capacitive sensors in [99,130,131].

The paper substrate has many advantageous features, e.g., it is inexpensive, eco-friendly, lightweight, expandable, foldable in three dimensions, and available everywhere [167,168,169,170,171]. Basak et al. [136] and Zhang et al. [152] developed their printed capacitive sensors on thermoplastic polyurethane (TPU) and aluminium substrates, respectively. Mikkonen et al. [140] proposed a novel approach using a water-soluble polyvinyl alcohol (PVA) layer as the substrate.

In this work, they reported a simple method for fabricating fully printed, elastomeric capacitive sensors with high flexibility and great sensitivity at a wide pressure range. A significant advantage of this work is the development of a straightforward method for inkjet printing of PDMS. They fabricated conductive mesh electrodes by inkjet printing silver nanoparticles on PVA, while a dielectric layer, consisting of a PVA mesh layer embedded in an inkjet-printed PDMS layer, was placed between the mesh electrodes to form a capacitive tactile pressure sensor.

Figure 6 shows some substrates used to manufacture capacitive sensors, while Figure 7 shows the capacitive tactile sensor manufactured in [140].

### 5.3. Piezoelectric

Among the existing piezoelectric materials, piezoelectric polymers are the most suitable for tactile sensor printing because of their superior flexibility and high transparency. These materials are being highly investigated due to their high mechanical stability and compatibility with solution-based printing technologies [172]. Polyvinylidene fluoride-triflouroethyleje, also known as P(VDF-trFE), is of particular interest, since it is a piezoelectric material commonly used to manufacture 3D printed [173,174] and inkjet-printed tactile sensors [82,134,175]. Two of its applications in the inkjet printing field are shown in Figure 8.

### 5.4. Other Transduction Methods

Different solutions based on the inkjet printing technique have been recently presented in [176,177], and they are shown in Figure 9. The work of Fu et al. [176] reported an all inkjet printing approach for fabricating flexible bimodal sensors (able to measure two parameters) and large-area sensing arrays, that can simultaneously sense bending strain and pressure. The sensor unit is composed of a bottom electrode on polyethylene naphthalate (PEN) and a top electrode on paper and silver nanoparticles (AgNPs) suspension was used as the conductive ink.

The device has excellent performance with low pressure-detection of 2 Pa and high stability over more than 5700 cycles for the pressure sensor. The device can recognize the radius of curvature of human activities and objects and can detect the pressure distribution and the curvature of irregular surfaces.

In the work of Salim et al. [177], an inkjet-printed kirigami-inspired split ring resonator (SRR) strain sensor is proposed, able to adapt to curved surfaces, easy to fabricate and cheap. Kirigami is an oriental technique of carving and folding of paper to obtain three-dimensional shapes starting from a single sheet, without removing pieces. Thus, in [177], two sheets of paper were used as the dielectric for compatibility with the kirigami technique, as shown in Figure 9b, and a conductive pattern was inkjet printed on the top paper using silver nanoparticle ink, whereas the ground plane on the bottom paper was inkjet printed using stretchable inks.

### 5.5. Features of the Proposed Manufacturing Solutions

Analysing the characteristics of the various manufacturing solutions, it is possible to understand how the inkjet printing technology can overcome the system-level integration challenges arising during tactile sensors design. First, the variety of flexible inks and substrate for various designs makes the sensors conformable to cover arbitrary curved robot body parts. Since the substrates are important for conformability, measurement principles, and applications, the common ones presented in this review article and their physical properties are listed in Table 1.

Furthermore, inkjet printing technology can be a good manufacturing solution as it permits a simple sensor layout design on specific surfaces. Then, in the inkjet printing process, the geometric placement techniques of sensors can be used during the design phase, once it is possible to create custom devices at a significantly lower cost compared with existing technologies. The possibility of designing custom-made sensors with accurate taxel placement may reduce the need of major geometric calibration procedures, thereby, reducing the setup time and costs of the sensor.

The expected repeatability of the inkjet printing techniques is also likely to reduce the extend of the calibration of the response of each tactile element. Finally, as mentioned in Section 4, the inkjet printing technology allows the placement of sensors far away from the processing unit in artificial skins for robot, introducing the possibility to print conductive wires on stretchable materials at very low-cost [148]. In Figure 10a, comparison between inkjet-printed wiring and conventional wiring is presented. The performances of the most relevant works are summarized in Table 2.

## 6. Inkjet Printed Tactile Sensors Applications

In the previous sections, we described the general features of inkjet-printed tactile sensor principles. From this standpoint, it possible to asses the most appropriate design solutions for different application areas. Tactile sensors have been used in different domains, including bioengineering (especially for areas, such as rehabilitation, soft robotics, wearable electronics, and surgical robotics) and industrial environments. In this section, we focus on the reported usage of inkjet-printed tactile sensors in these sectors, that are finally shown in Figure 11.

Inkjet printing technology has an immediate advantage for large area applications (as for bioengineering and industrial domains). A future development for this technology can be investigated for applications regarding the realization of miniaturized sensors. The latter is still an open problem for this type of technology, and it will be discussed in the next section.

### 6.1. Tactile Sensing for Rehabilitation

Rehabilitation is one of the main fields of application for tactile sensing. In particular, robot-assisted rehabilitation has been widely used for repetitive and interactive treatments, such as upper limbs orientation and recovery [181]. Among the various technologies, smart tactile gloves have been largely used, especially in stroke rehabilitation [181]. Inkjet printing is an emerging technique for the patterning of sensors and conducting traces onto gloves or textiles [72,182,183,184,185]. Tactile sensors can also be printed onto the glove to measure the pressure exerted by the fingers. Ervasti et al. [166] proposed a stretchable inkjet-printed piezoresistive tactile sensor with high sensitivity to bending for finger joints flexion measurements.

### 6.2. Tactile Sensing for Soft Robotics

Soft robotics is a subfield of robotics that concerns the design, control, and fabrication of robots composed of compliant materials instead of rigid links. The applications of soft robots is gaining a growing relevance, as they are flexible and adaptable to operate in complex environments. Tactile sensors are one of the most important components for soft robotics for their role in the perception of the physical interaction with the environment [186].

Inkjet printing technology emerged in this domain for its potential to manufacture highly integrated and custumized sensors, as well as for the possibility of designing small devices. The development of tactile sensors for soft robots evolves with the investigation and development of inkjet-printed electronics solutions for deformable and stretchable materials [166,187,188].

### 6.3. Tactile Sensing for Wearable Devices

Wearable devices sensorized with tactile sensors have been proposed in the literature [189]. Currently, several monitoring devices face drawbacks, such as rigid equipment, low accuracy and high expenses. Inkjet printing technology leads to tactile sensors with a high level of wearability in terms of bendability and flexibility, enhancing their interfaces with the contact objects. Therefore, printed tactile sensors are capable of conformably covering arbitrary curved surfaces over their stiff counterparts without incurring damage [189].

By making electronic devices flexible, it is possible to significantly enhance their healthcare applications in the emerging Internet of Things (IoT) [190]. Furthermore, wearable electronic skin (e-skin) devices with skin-like properties can adhere conformally to the skin and interface with the human body to mimic the human somatosensory system and enable convenient and non-invasive tracking of physical and biochemical signals [166,185,189,191,192,193].

The most promising tactile sensing applications are gait detection and analysis [194] and motion monitoring [179]. Furthermore, as remarked in [86], printed tactile sensors can play a major role in applications as expression recognition, diagnosing conditions, such as Parkinson’s, sign language translation and body gesture.

As mentioned in the previous sections, Albrecht et al. [148] proved the development of inkjet-printed stretchable wires, used to connect sensors with the remote processing unit, integrated in smart clothes, or, more in general, for artificial skin for robots. This innovative manufacturing solution is useful for tactile sensors in sportswear, compression underwear, and in robotic applications [148].

### 6.4. Tactile Sensing for Surgical Robotics

Surgery is a developing area where tactile sensing is of central importance. Minimally invasive surgery (MIS) is now routinely used as the preferred choice for many operations [6]. Although the benefits of MIS technology have been proven, the limitations of two-dimensional visualization, lack of haptic feedback, and long learning times are their limiting factors [4]. In this context, tactile sensing is used to study the interactions between the confined tissue environments and the instruments based on contact information and haptic feedbacks [132,195,196,197,198,199].

Force sensing capability in an instrument helps to improve the instrument-body interaction, and it can reduce damages in order to ensure effective manipulation. Thus, for surgical and medical devices, sensors need to be compact, low power consuming and less bulky with simple associated electronics [200]. The inkjet printing technology could bring improvements in the development of these devices leading to soft and flexible sensors that can also ensure better compatibility with human tissue.

The first example is given by Ponraj et al. [132], who described the design, fabrication, and application of a soft inkjet-printed piezoresistive tactile sensor used for safe human–tissue interactions. The sensor can be integrated with surgical devices providing forces feedback. A key contribution of the paper is to prove inkjet printing as a cheap solution for the sensorization of surgical equipment.

### 6.5. Tactile Sensing for Industrial Applications

Within the industry 4.0 paradigm, advanced robotic technologies are a key element to enable safe human–robot interactions. Robots and manipulators are expected to safely adapt their behaviour with respect to obstacles and humans [201]. Human operators become a fundamental part of a robot’s operating environment, since they represent external unpredictable agents to comply with.

Thus, new technologies are required to implement and promote safe and proficient human–robot interactions. Recently, 3D and inkjet printing for tactile sensing have been used in this context [129,178,202]. The capability to develop inkjet-printed sensors on large area, opens the possibility of fully sensing robots that perceive their surroundings in much greater detail, ensuring safer interactions between humans and robots.

## 7. Open Problems

This survey aims to provide an overview of how inkjet printing can boost the development of tactile sensors on a large scale of production, leading to high *technology readiness level (TRL)* cost effective devices ready for the market. As discussed in Section 2, the fundamental requirements for the design of large-area tactile sensors are conformability, distribution and placement, spatial calibration, and routing of wires. The difficulties in obtaining these conditions have been addressed with inkjet printing technology.

Although inkjet printing can be considered one of the most promising technologies in tactile sensors scenario, some open questions have not been addressed in the literature with this technology yet. In particular, open problems for the development of complex tactile sensors refer to vias fabrication. A relevant problem is the robust bonding of chips on the sensor substrate, which is also related to the flexibility requirements of the skin. Moreover, the ageing of the materials used, in particular of the inks, which might limit the operational and/or shelf life of the devices.

Finally, a major challenge is the possibility of a direct printing of sensors on the target surfaces by exploiting the state of the art of robotic technologies. The problem of vias fabrication is critical, especially in multilayered tactile devices, in which functional devices from different layers need to be connected together by vertical interconnects [61].

Another challenge is to obtain the integration of the chip with the substrate, guaranteeing the mechanical stability of the system, reliability, and robustness, avoiding noise in the acquired signals. Even if sensors built with traditional technologies may have the electronics mounted on the device, the problem is in the bonding on inkjet-printed tactile sensors, whose robustness can be lower than those of a traditional devices.

Some relevant upgrades were proposed in [203]. Furthermore, one of the major problems that affects the tactile sensors’ functioning is the ageing of the materials, which change their sensitivity over time. The challenge is the study of new materials that are not affected by ageing and that they can be printed by inkjet printing. Finally, the direct inkjet printing of sensors on the target surface remains a challenge, and it may require other forms of additive manufacturing [204].

The possibility of direct printing allows for an understanding of how exactly the sensor is positioned on the robot body, avoiding spatial calibration procedures. Inspired by the idea of six-axis additive manufacturing [205,206], inkjet printing technology could find future application in the production of miniaturized devices, thus, allowing the creation of patterns for transducers even on small and non-planar surfaces.

Table 3 systematizes, in detail, all the aforementioned open problems.

## 8. Conclusions

In this review, we presented many papers to demonstrate how inkjet printing technology can replace the traditional fabrication techniques used for tactile sensors. An overview of the major requirements and challenges arising in the large-area tactile sensors field was introduced, followed by the fundamental transduction principles of tactile sensors. Then, a general description of the printing process, with its advantages and shortcomings, was described, highlighting the typical parameters with relevance to tactile sensors that can be achieved with inkjet printing technology.

This review also covered the challenges of inkjet-printed tactile sensors: problems that can arise during the manufacturing processes were discussed, introducing possible solutions. Furthermore, the state of the art regarding inkjet-printed tactile sensors is provided, introducing an assessment of the different characteristics of the fundamental transduction principles and demonstrating how inkjet printing can overcome the system-level integration challenges. With a complete vision of the inkjet-printed tactile sensors domain, the paper contributes to assessing the most appropriate design solutions for different application areas.

Finally, the survey provides a view of the most relevant open problems that should be addressed to improve the effectiveness of these processes. The survey particularly focuses on the inkjet printing of large-area tactile sensors for robotics applications; however, all the treated topics can be extended to generic systems.

## Figures and Tables

**Figure 1 sensors-22-02332-f001:**
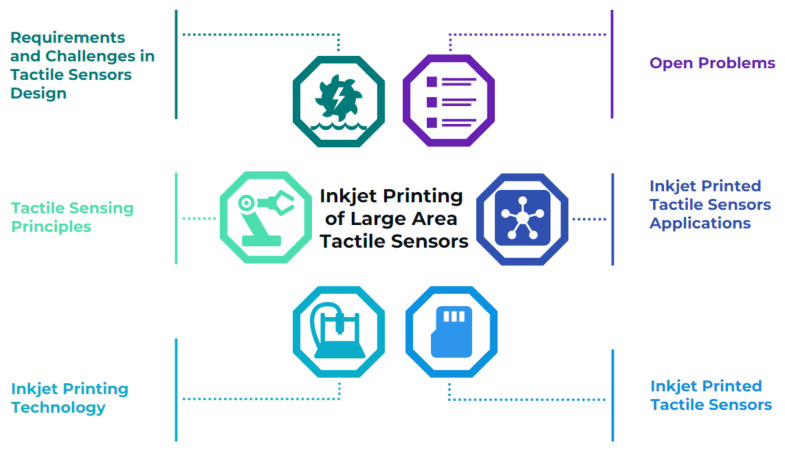
Map of all treated topics.

**Figure 2 sensors-22-02332-f002:**
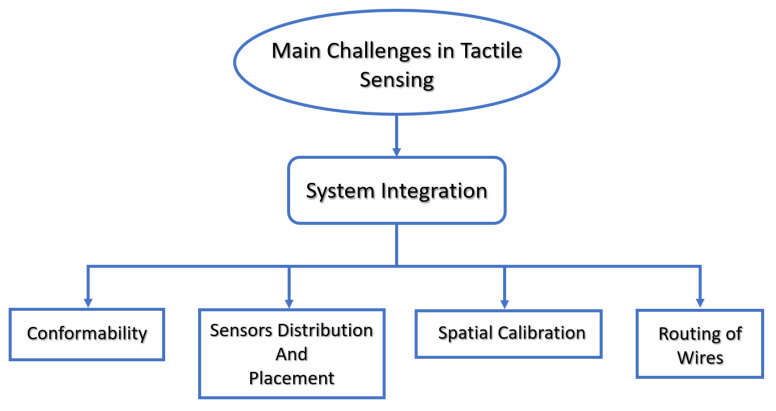
The main challenges in tactile sensing addressed with the inkjet printing technology.

**Figure 3 sensors-22-02332-f003:**
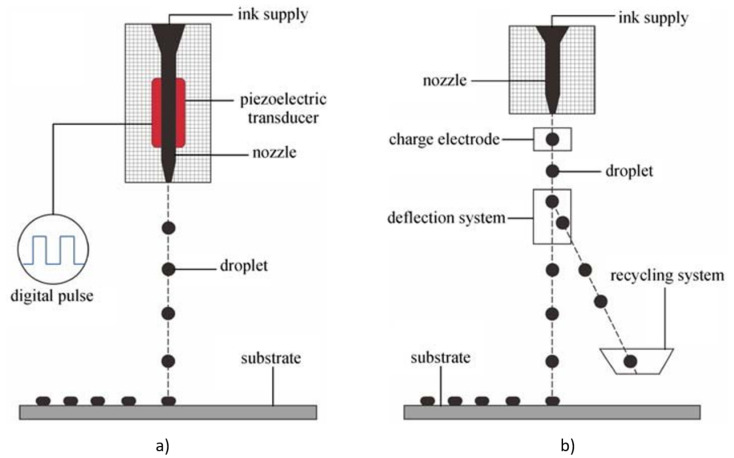
Schematic diagram of the inkjet printing processes. (**a**) DoD mode. (**b**) Continuous mode.

**Figure 4 sensors-22-02332-f004:**
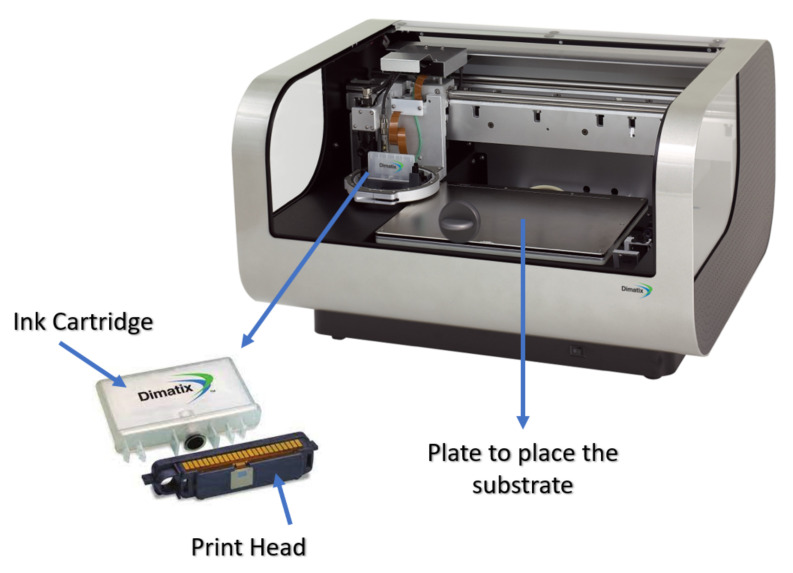
Main components in inkjet printing processes: inkjet printer, ink, and substrate.

**Figure 5 sensors-22-02332-f005:**
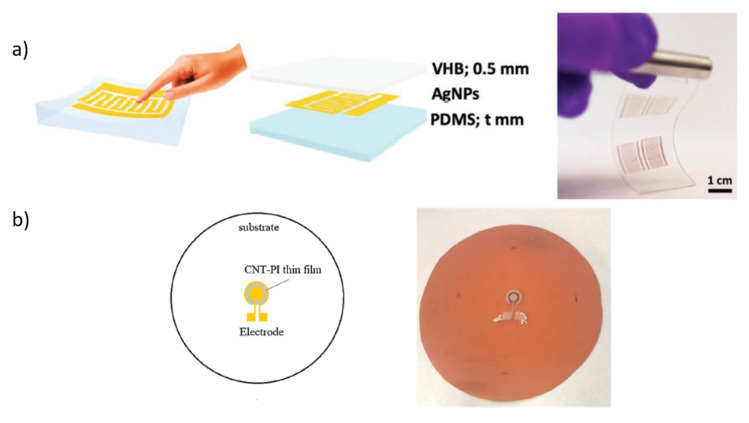
Schematic illustrations of printed piezoresistive sensors: (**a**) [139] and (**b**) [101].

**Figure 6 sensors-22-02332-f006:**
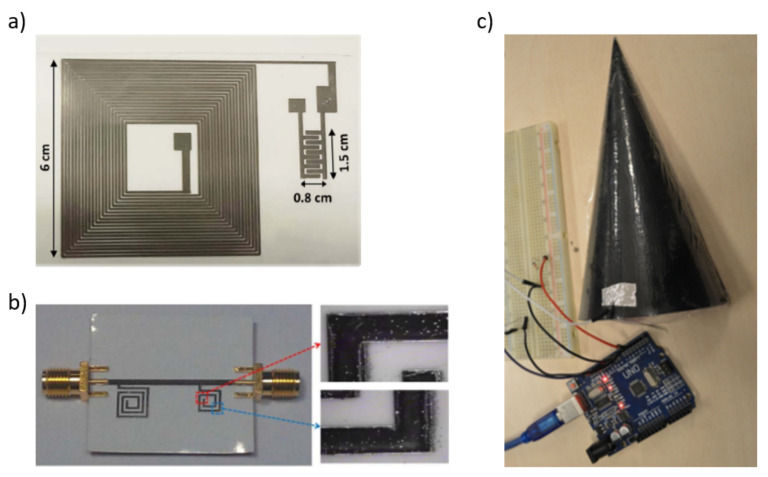
Inkjet-printed capacitive sensor layout on (**a**) PET substrate [129]. (**b**) Photopaper [130]. (**c**) TPU substrate [136].

**Figure 7 sensors-22-02332-f007:**
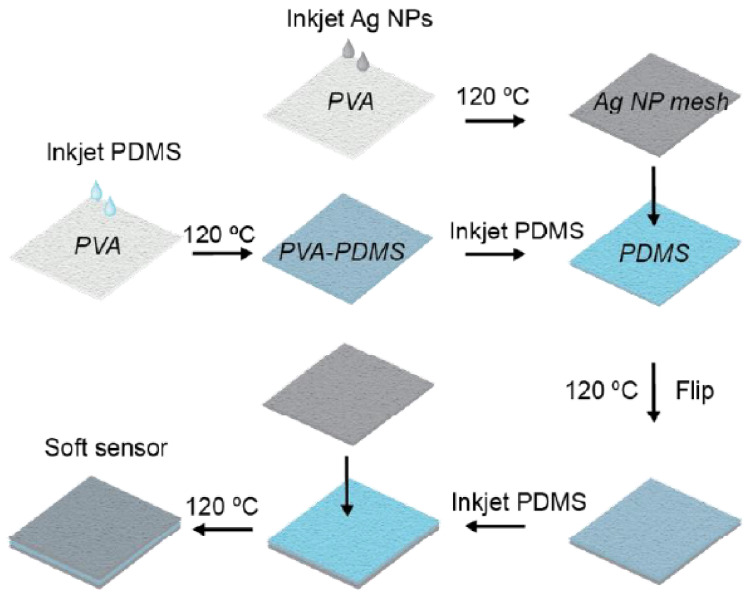
Schematic illustration of the sensor manufacturing process in [140].

**Figure 8 sensors-22-02332-f008:**
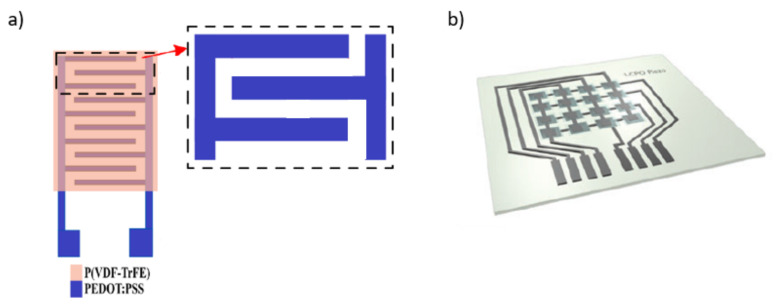
Schematic illustration of printed piezoelectric sensors: (**a**) [134] and (**b**) [175].

**Figure 9 sensors-22-02332-f009:**
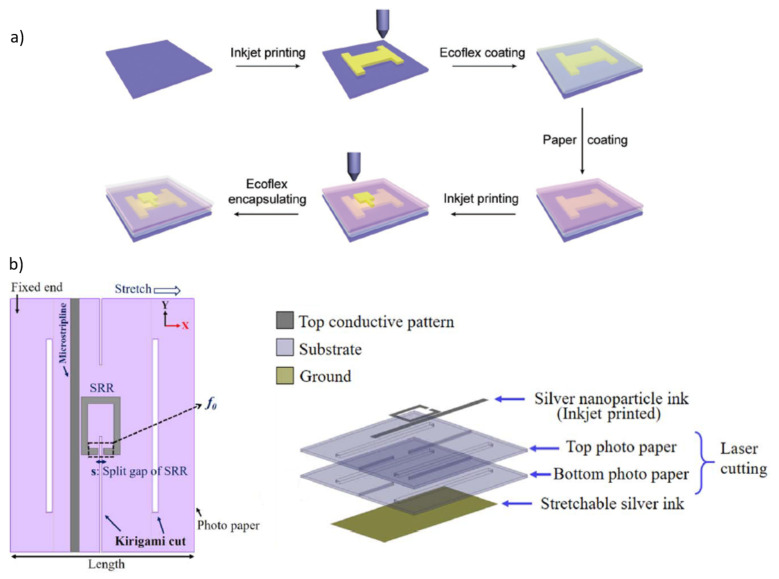
Schematic illustrations of sensors manufacturing processes in [176,177]: (**a**) bimodal sensor and circuit diagram in [176], (**b**) sensing principle and layout for the proposed inkjet-printed kirigami-inspired split ring resonator (SRR) strain sensor at zero strain and corresponding resonance frequency ($f_{0}$) [177].

**Figure 10 sensors-22-02332-f010:**
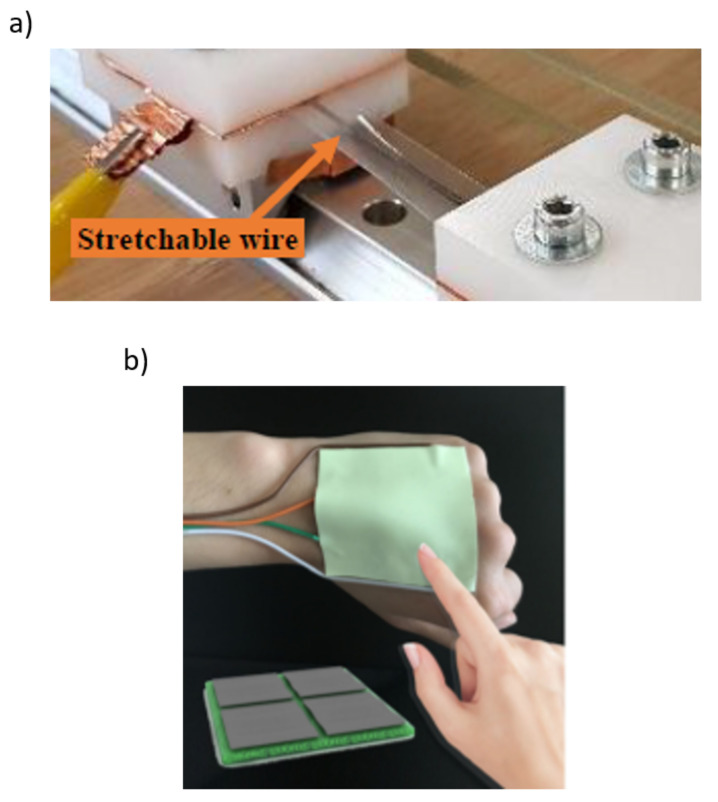
Comparison between (**a**) inkjet-printed wiring [148] and (**b**) conventional wiring [180].

**Figure 11 sensors-22-02332-f011:**
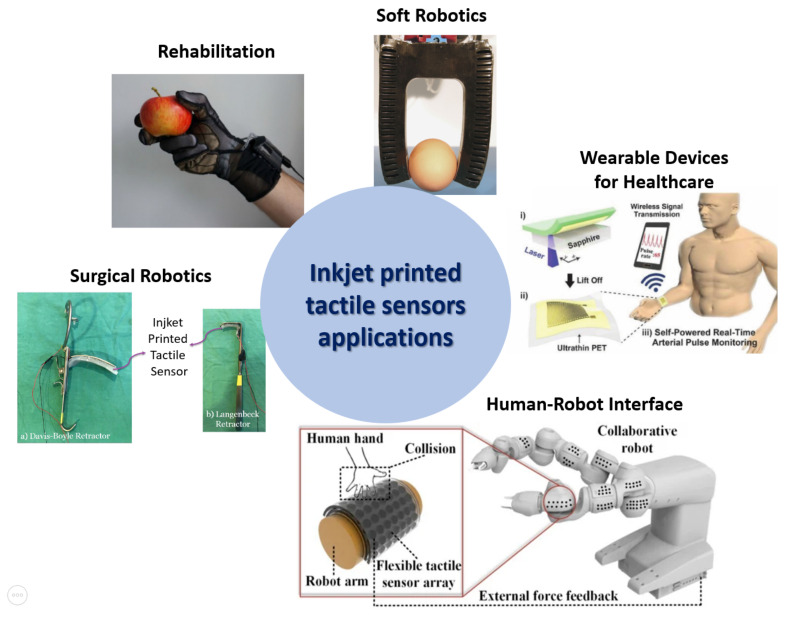
Diagram of inkjet-printed tactile sensor relevant applications: rehabilitation [185], soft robotics [188], wearable devices [192], surgical robotics [132], and human–robot interface [178].

**Table 1 sensors-22-02332-t001:** Substrates and their physical properties as presented in this review article.

Substrate	Thickness [µm]	Transparency [%]	Density [g/cm${}^{3}$]	Young Modulus [GPa]	Ref.#
PAPER	25–250	-	0.6 - 1.0	0.5–3.5	[99,130,131,166]
POLYMIDE	12–125	-	1.4	2.5	[101,126]
PET	16–100	90	1.38	2.8	[100,129,132,137,142]
PEN	12–250	87	1.4	3.0	[128,144,175,176]
PDMS	5–1500	92	0.965	0.57–3.7	[98,139,145,146,147,148,177,178,179]

**Table 2 sensors-22-02332-t002:** Performances of the most relevant works presented in this review article.

Principle	Sensitivity	Detection Range	Repeatability	Ref.#
Piezoresistive	0.48 kPa${}^{-1}$	15 kPa	High (1000 cycles)	[139]
	-	151 kPa	Good	[101]
Cacapitive	4 MPa${}^{-1}$	50 kPa	High (2000 cycles)	[140]
Piezoelectric	3.9 ± 0.5 pC/N	-	Good	[134]
Others	-	1000 kPa	High (4500 cycles)	[176]
	4.2 × 10${}^{7}$ Hz/%	-	Good	[177]

**Table 3 sensors-22-02332-t003:** Open problem descriptions.

Open Problems	Description
Vertical vias fabrication	The fabrication of vias is a critical problem, especially in multilayered tactile devices, in which functional devices from different layers need to be connected together by vertical interconnects.
Bonding of chips on the sensor substrate	Since tactile sensors are devices subjected to significant mechanical stresses due to contacts, the problem of bonding and of sensor integration with the electronics is a problem from the durability and robustness point of view. This problem regards in particular the inkjet-printed tactile sensors, whose robustness can be lower than those of a traditional devices.
Ageing of materials	The ageing of the material used is a problem that has to be considered, as it leads to a decrease in sensor lifetime and sensitivity.
Direct printing of sensors on the target surface	The direct inkjet printing of sensors on the target surface is still a challenge, and it may requires other forms of additive manufacturing. The possibility of direct printing allows understanding how exactly the sensor is positioned on the robot body, avoiding spatial calibration procedures.

## Data Availability

Not applicable.

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
