# Peer review of "An Atlas for the Inkjet Printing of Large-Area Tactile Sensors"

_sensors, 2022, doi:10.3390/s22062332_

Round 1

Reviewer 1 Report

The Review gives an overview about inkjet-printed tactile sensors and the five main fields of applications. There are already different reviews on inkjet-printed sensors for wearables as well as tactile sensors using flexible electronics, but this overview was not published before. The authors have performed a thorough literature review presented in the extensive number of references. Thus, the paper is of interest to the reader. The English language is mostly appropriate and understandable, though often singular and plural of object and verb are disarranged. The main shortcoming is the overall structuring and presentation of some figures and tables. see attached.

Reviewer 2 Report

This paper reviews the inkjet printing technique for fabrication of large area tactile sensors. The paper introduces recent manufacturing techniques and design issues in inkjet printing methodology.

Although this review paper focuses the fabrication issues, the statements of signal processing and sensing performances are indispensible in introduction of various tactile sensors, especially inkjet printed sensors.

The signal processing issues in each type and sensing performances(dynamic range, sensitivity, etc.) should be addressed and discussed for comprehensive point of view.

Also grammatical errors and misspellings were founded in many sentences. The English proofreading is recommended.

Reviewer 3 Report

Comment for Authors

  1. A summary of inkjet-based sensor design solutions should be described in tables to show the differences in characteristics, fabrication, materials, functions, advantages, disadvantages, and so on such as:
    1. Features of the additive manufacturing techniques
    2. Various types of printing methods of inkjet-based tactile sensors
    3. The presentative fabrication steps
    4. Fabricated materials and sensor features
    5. Properties of a few commonly used substrate materials
    6. The promising applications of inkjet-based tactile sensors
    7. Representative tactile sensors and their applications…...
    8. ...
  2. Piezoresistive, capacitive piezoelectric other transduction methods, should also tabulate to compare specifications.
  3. The realization of stretchable devices on the real robot application should be described more clearly. The application of each sensor for each type of robot should be addressed clearly in tabular.
  4. Has not inductive tactile sensing been mentioned in this paper?
  5. There have been many review articles on inkjet-base tactile sensing, what points are important in this article when comparing them to other review articles?
  6. This article lacks visualization.

Round 2

Reviewer 1 Report

You have thoroughly implemented the comments and revised the manuscript. The quality and content of the paper has been improved significantly. It is now a good and valuable contribution to the investigated field of research. The English language still needs some rework, especially in the new paragraphs.

Reviewer 3 Report

After reading and reviewing the author's revision, the reviewer agrees with the author's revision. The author has revised according to the comments of the reviewer carefully and with enthusiasm.  The scientific contribution of the paper is substantial.

The author has added the open problems section.
The reviewer has an additional suggestion for the author that the open problems section can be systematized into a detailed table which will be an important contribution of the paper that has not been published in other review papers. 
Tables 1 and 2 do not fully list the necessary information. Further clarification and detail are needed for these tables by referring to other review articles listed in this reference section. 
